# Inverse association of dietary consumption of n3 and n6 fatty acids with hyperuricemia among adults

Huakai Wang[1☯], Chao Zhang[2☯], Yuxin Sun[3☯], Sirui Sun[4], Zhe Wang[1]*, Honggang Xiang[1]*

1 Department of General Surgery, Pudong New Area People's Hospital, Shanghai, China, 2 Qingdao Medical College, Qingdao University, Qingdao , China, 3 . Department of Oncology, Xinhua Hospital Affiliated to Shanghai Jiao Tong University School of Medicine, Shanghai, China, 4 The Hockaday School, Dallas, Texas, United States of America

☯ Huakai Wang, Chao Zhang and Yuxin Sun contributed equally to this work and should be considered co-first authors.
* honggangxiang@outlook.com (HX); wangzhe@shpdph.com (ZW)

## Abstract

### Background

The precise link between dietary consumption of n-3 and n-6 fatty acids and hyperuricemia remains equivocal. Accordingly, the purpose of the current investigation is to clarify any possible associations between the consumption of n-3 and n-6 fatty acids and hyperuricemia in the context of American adults.

### Materials and methods

The present investigation employed a cross-sectional design, comprising a sample of 21,120 American adults above the age of 20 from the National Health and Nutrition Examination Survey (NHANES) waves between 2007 and 2016. The dietary consumption of n-3 and n-6 fatty acids was measured through two 24-h dietary recall interviews. To assess the relationships of dietary consumption of n3 and n6 fatty acids with hyperuricemia, we applied multivariable logistic regression, t tests, chi-square tests, and restricted cubic spline. To determine the robustness of our findings, sensitivity analyses were also carried out.

### Results

The results of the multivariable logistic regression models indicated a significant correlation between dietary consumption of n3 and n6 fatty acids and hyperuricemia. The ORs with 95% CIs of hyperuricemia for the highest tertile versus lowest tertile of dietary consumption of n3 and n6 fatty acids were 0.76 (0.66, 0.88) (p < 0.001) and 0.72 (0.64, 0.82) (p < 0.001), respectively. Moreover, dose–response analyses revealed a linear relationship between n-3 and n-6 fatty acid consumption and the risk of hyperuricemia.

**Data availability statement:** All relevant data are available in National Health and Nutrition Examination Survey database: https://www.cdc.gov/nchs/nhanes/?CDC_AAref_Val= https://www.cdc.gov/nchs/nhanes/index.htm

**Funding:** This work was supported by National Natural Science Foundation of China (NO.82103390 to HW), Technology Development Project of Pudong Science, Technology and Economic Commission of Shanghai (Grant No.PKJ2020-Y39 to HX) and Outstanding Leaders Training Program of Pudong Health Commission of Shanghai (No.PWR12023-03 to HX).

**Competing interests:** The authors have declared that no competing interests exist.

## Conclusion

The findings of this study indicate a significant inverse correlation between the dietary consumption of n3 and n6 fatty acids and hyperuricemia in the US adult population. Notably, there was no significant relationship between the n6:n3 ratio and hyperuricemia.

## Introduction

Uric acid is the end-product of purine metabolism, and an elevation in serum uric acid levels beyond the normal range results in hyperuricemia. Extant research has highlighted the association between hyperuricemia and heightened risks of gout, as well as several other deleterious conditions, such as cardiovascular disease, metabolic syndrome, chronic kidney disease, type 2 diabetes, and lipid metabolism disorders [1–3]. Presently, hyperuricemia is emerging as a significant public health challenge, with epidemiological data revealing an escalating trend in the incidence of hyperuricemia and gout in Western countries [1,4]. Consequently, there is a pressing need to explore modifiable risk factors associated with hyperuricemia to manage serum uric acid concentrations.

N-3 fatty acids represent polyunsaturated fatty acids that have the first double bond at the third carbon atom from the methyl end, while N-6 fatty acids have the first double bond at the sixth carbon atom [5]. Linoleic acid (LA) (n-6) and α-linolenic acid (ALA) (n-3) are considered essential fatty acids since they cannot be produced by humans or other higher animals [6]. Previous studies have reported a correlation between dietary consumption of n3 and n6 fatty acids and the possibility of developing metabolic syndrome [7–9], cardiovascular diseases [10], obesity, cancer [7,11], type 2 diabetes and gestational diabetes [12–14]. Meanwhile, several studies indicated that n3 and n6 fatty acids might decrease the level of serum uric acid [15,16], while another study showed no relationship between n3, n6 and serum uric acid levels [17]. In addition, no studies reported a dose–response relationship.

Given the inconsistencies in previous research findings, we made use of data from the National Health and Nutrition Examination Survey (NHANES) from 2007 to 2016 in this investigation to explore the relationships between the consumption of n3 and n6 fatty acids, as well as n6:n3 ratios, and the risk of hyperuricemia.

## Materials and methods

### Study population

NHANES is a survey program initiated by the Centers for Disease Control and Prevention of America that gathers detailed information on the population's nutritional status and health twice a year. NHANES uses a complex multistage stratified probability cluster sampling technique to obtain a representative sample of noninstitutionalized civilians. The NHANES database is accessible to everyone in the world. The National Center for Health Statistics Research Ethics Review Board approved the current investigation, and all subjects provided written informed consent [5].

In this study, we analyzed publicly available data from five successive cycles of NHANES (2007-2008, 2009-2010, 2011-2012, 2013-2014, 2015-2016). We first identified 50588 participants in the database and subsequently developed the following exclusion criteria: a) younger than 20 years old, n = 21387; b) lack of complete uric acid reading, n = 2846; c) unreliable 24-h dietary recall data, n = 4756; d) missing weight data, n = 171; e) pregnant or lactating, n = 202; f) females with total energy intake < 500 or > 5000 kcal/day and males with < 500 or > 8000 kcal/day. Ultimately, this study enrolled 21120 adult participants (Fig 1).

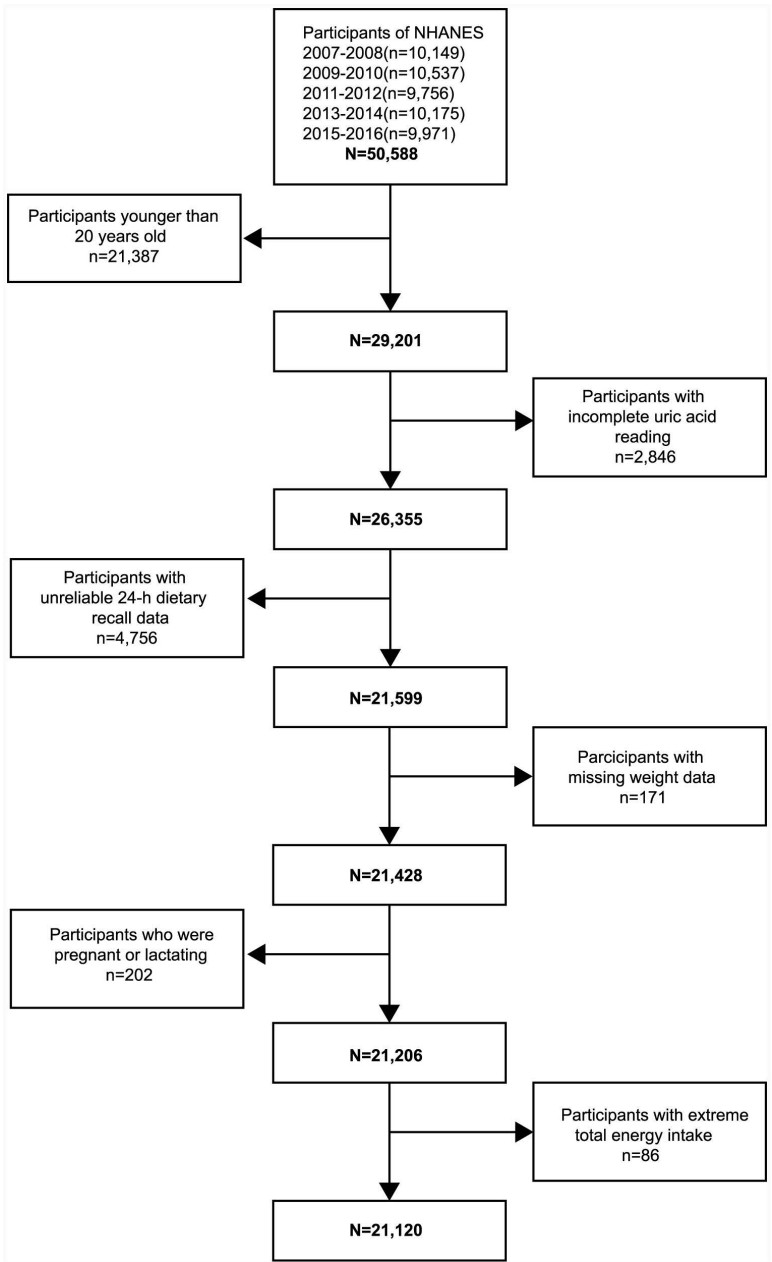

**Fig 1. Flow chart of the screening process for the selection of eligible participants.**

## Serum uric acid measurement and definitions of hyperuricemia

A Beckman UniCel® DxC800 Synchron or a Beckman Synchron LX20 (Beckman Coulter, Inc., Brea, CA, USA) was used to measure uric acid concentrations in serum, plasma, or urine with a timed endpoint approach. Allantoin and hydrogen peroxide are created when uricase oxidizes uric acid, and the latter is then used in a reaction catalyzed by peroxidase with 4-aminoantipyrine (4-AAP) and 3,5-dichloro-2-hydroxybenzene sulfonate (DCHBS). The system monitors changes in absorbance at 520 nm at predetermined intervals to determine whether a colored product has been produced by this reaction. The amount of uric acid in the

sample directly correlates to the change in absorbance. Serum uric acid levels of > 6.0 mg/dL for women and > 7.0 mg/dL for men were defined as hyperuricemia [18].

## The dietary consumption of n3 and n6 fatty acids

N3 and N6 fatty acid intake data were gathered from two 24-hour dietary recall interviews [19]. The interviews took place face-to-face at the mobile examination center (MEC) and then over the phone three to ten days later. Alpha-linolenic acid (ALA), stearidonic acid (SDA), docosapentaenoic acid (DPA), eicosapentaenoic acid (EPA), and docosahexaenoic acid (DHA) are all members of the n3 fatty acid group. Linoleic acid (18:2) and arachidonic acid (20:4) are both members of the n6 fatty acid group. Based on the Dietary Research Food and Nutrition Database for Dietary Studies maintained by the U.S. Department of Agriculture, the average daily consumption of n3 and n6 fatty acids was computed and weighed. Tertiles were created based on dietary n3 and n6 fatty acid consumption as well as the n6:n3 ratio.

## Covariates

To account for potential confounders, a number of covariates were chosen based on previous research. Potential confounders in the demographic variables are as follows: a) age (20-40 years, 40-60 years, and ≥ 60 years), b) sex (male and female), c) ethnicity (Mexican American, Other Hispanic, Non-Hispanic White, Non-Hispanic Black, and Other ethnicities), d) educational level (below high school, high school, and above high school), e) marital status (married/living with partner and widowed/divorced/separated/never married), f) annual household income (<$20,000 and ≥ $20,000), g) body mass index ( < 25.0 kg/m2, 25.0 to 30.0 kg/m2 and ≥ 30.0 kg/m2), h) hypertension status (yes/no), i) smoking status (ever smoked ≥ 100 cigarettes in life or not), j) drinking status (having at least 12 alcohol drinks per year or not), k) work activity (vigorous activity, moderate activity, and other), l) recreational activity (vigorous activity, moderate activity, and other), m) diabetes (yes/no), and n) total cholesterol (TC) and high-density lipoprotein cholesterol (HDL-C). Self-reported diabetes history was used to define diabetes. Hypertension was defined as the use of antihypertensive medications, a mean systolic blood pressure > 130 mmHg or a mean diastolic blood pressure < 80 mmHg.

## Statistical analysis

The normality of continuous variables was examined with Kolmogorov–Smirnov tests [20]. The mean ± standard deviation was used to express normally distributed variables. The differences between the hyperuricemia group and the nonhyperuricemia group were compared using t tests or chi-square testing depending on the characteristics of the variables. Tertiles were used to group the adjusted dietary consumption of n3 and n6 fatty acids and the n6:n3 ratio. We set tertile 1 as the reference group. The connection of the consumption of n3 and n6 fatty acids as well as the n6:n3 ratio with hyperuricemia was examined using multivariable logistic regression models. For ethnicity, sex, and age, Model 1 was modified. Additional adjustments were made in Model 2 for factors such as education level, marital status, annual household income, BMI, hypertension, drinking and smoking habits, work and recreational activity, diabetes, TC, HDL-C, and TG. To further explore the relationship between dietary n3 and n6 fatty acid intake and the risk of hyperuricemia, we performed stratified analyses by sex and age. With three knots placed at the 5th, 50th, and 95th percentiles of the exposure distribution, we used restricted cubic splines (RCS) to analyze the dose–response relationship. The confounding factors that were addressed in multivariate-adjusted Model 2 were also adjusted in the restricted cubic spline. We modeled the median values of each tertile as a continuous variable in the overall models to examine the trend across tertiles of daily dietary consumption

of n3 and n6 fatty acids and the n6:n3 ratio. Because kidney diseases and obesity may affect the result [21], we further conducted sensitivity analyses to estimate the robustness of our study by excluding participants with obesity (BMI ≥ 30 kg/m²) and significant renal dysfunction (eGFR lower than 60 mL/min/1.73 m²). Testing the value of the coefficient for zero of the second spline resulted in the calculation of the p value for nonlinearity. In our investigations, a nationally representative estimate was produced using the appropriate sampling weights, primary sampling unit, and stratum information. Stata 14.0 (Stata Corporation, College Station, Texas, USA) was used to conduct each statistical analysis. Statistical significance was defined as a two-sided $p < 0.05$.

### Ethics approval and consent to participate

The Ethics Review Board of the National Center for Health Statistics approved all NHANES protocols.

## Results

### Baseline characteristics

The characteristics of NHANES participants with hyperuricemia are presented in Table 1. Among 21,120 participants, the prevalence of hyperuricemia was 19.5%. Compared with the nonhyperuricemia group, people with hyperuricemia tended to be male, have non-Hispanic white and black ethnicity, have lower household income and educational level, have age more than 60 years old, have diabetes and hypertension, live alone, smoke more, have lower recreation activity, and have higher BMI, TG, and TC. In addition, the total adjusted consumption of n3 and n6 fatty acids in hyperuricemia was significantly lower in participants without hyperuricemia.

### Associations between dietary consumption of n3 and n6 fatty acids and hyperuricemia

Table 2 displays odds ratios (ORs) and 95% confidence intervals (CIs) for hyperuricemia based on tertiles of adjusted dietary consumption of n3 and n6 fatty acids and the n6:n3 ratio. Compared with the lowest tertile, the highest tertile of dietary consumption of n3 and n6 fatty acids was linked to a lower prevalence of hyperuricemia in the binary logistic regression analysis, with ORs of 0.50 (95% CI: 0.45, 0.56) and 0.49 (95% CI: 0.45, 0.54), respectively. However, no significant association was found between the n6:n3 ratio and the risk of hyperuricemia in any model. After adjusting for age, sex, and ethnicity (Model 1), dietary consumption of n3 and n6 fatty acids remained significantly associated with a decreased prevalence of hyperuricemia. Furthermore, after additional adjustments for educational level, marital status, annual household income, hypertension, smoking status, work activity, recreational activity, diabetes, BMI, TC, HDL-C, and TG (Model 2), the ORs (95% CIs) for hyperuricemia for dietary consumption of n-3 and n-6 fatty acids were 0.76 (0.66, 0.88) and 0.72 (0.64, 0.82), respectively.

### Consumption of N3 and n6 fatty acids stratified by age

Table 3 presents the associations between dietary consumption of n3 and n6 fatty acids and hyperuricemia risk stratified by age. After adjusting for age, sex, ethnicity, educational level, marital status, annual household income, hypertension, smoking status, work activity, recreational activity, diabetes, BMI, TC, HDL-C, and TGs (Model 2), the ORs (95% CIs) of hyperuricemia among participants aged 20 to 40 were 0.61 (0.48, 0.77) for dietary n3 fatty acid intake and 0.68 (0.56, 0.84) for dietary n6 fatty acid intake. For those aged 40-60 years old,

**Table 1. Characteristics of National Health and Nutrition Examination Survey (NHANES) participants by hyperuricemia.**

| | Non-Hyperuricemia | Hyperuricemia | P Value |
|---|---|---|---|
| Number of Participants (%) | 16,820(80.5) | 4,300 (19.5) | |
| Age group (%)[1] | | | <0.01 |
| 20-40 years | 5,721 (35.4) | 1,035 (27.75) | |
| 40-60 years | 5,891 (39.6) | 1,305 (35.99) | |
| ≥60 years | 5,208 (25.0) | 1,960 (36.25) | |
| Gender (%)[1] | | | <0.01 |
| Male | 7906(46.3) | 2316(55.5) | |
| Female | 8914(53.7) | 1984(44.5) | |
| Ethnicity (%)[1] | | | <0.01 |
| Non-Hispanic White | 7360(68.8) | 2072(72.6) | |
| Non-Hispanic Black | 3213(10.0) | 1047(11.8) | |
| Mexican American | 2693(8.6) | 467(5.8) | |
| Other Hispanic | 1897(5.8) | 327(3.7) | |
| Other ethnicities | 1657(6.8) | 387(6.1) | |
| Educational level (%)[1] | | | |
| Below high school | 3973(15.5) | 1020(15.9) | 0.0308 |
| High school | 3759(21.6) | 1047(24.2) | |
| Above high school | 9074(62.9) | 2229(62.3) | |
| Marital status (%)[1] | | | <0.01 |
| Married/Living with partner | 10,243(65.3) | 2,457(62.1) | |
| Widowed/Divorced/Separated/Never married | 6,571(34.7) | 1,843(37.9) | |
| Household income (%)[1] | | | <0.01 |
| <$20,000 | 3,245(13.2) | 987(15.4) | |
| ≥$20,000 | 12,854(86.8) | 3,141(84.6) | |
| Body mass index (%)[1] | | | <0.01 |
| <25 kg/m² | 5332(33.2) | 569(12.7) | |
| 25 to <30 kg/m² | 5776(34.6) | 1236(29.3) | |
| ≥30 kg/m² | 5688(32.2) | 2483(58.1) | |
| Hypertension (%)[1] | | | <0.01 |
| Yes | 8884(58.0) | 2935(64.6) | |
| No | 7658(42.0) | 1317(35.4) | |
| Smoking at least 100 cigarettes in life (%)[1] | | | <0.01 |
| Yes | 7322(46.6) | 2084(48.1) | |
| No | 9490(56.4) | 2214(51.9) | |
| Have at least 12 alcohol drinks/year (%)[1] | | | 0.631 |
| Yes | 11,492(77.6) | 2,985(77.2) | |
| No | 4404(22.4) | 1145(22.8) | |
| Recreation activity (%)[1] | | | <0.01 |
| Vigorous activity | 3858(27.2) | 672(18.7) | |
| Moderate activity | 4515(28.7) | 1151(29.8) | |
| Other | 8447(44.1) | 2477(51.6) | |
| Work activity (%)[1] | | | 0.370 |
| Vigorous activity | 3172(20.5) | 808(20.6) | |
| Moderate activity | 3641(23.6) | 930(25.0) | |
| Other | 15,930(55.9) | 2562(54.4) | |
| Diabetes (%)[1] | | | <0.01 |

*(Continued)*

**Table 1.** (Continued)

|  | Non-Hyperuricemia | Hyperuricemia | P Value |
|---|---|---|---|
| Yes | 1904(8.5) | 794(14.8) |  |
| No | 14559(91.5) | 3366(85.2) |  |
| Total energy intake (kcal/day)[2] | 2038.5(801.0) | 1971.0(786.3) | <0.01 |
| TC (mg/dL)[2] | 192.4(41.1) | 197.4(43.2) | <0.01 |
| HDL-C (mg/dL)[2] | 53.8(16.1) | 48.9(15.3) | <0.01 |
| TG (mg/dL)[2] | 148.8(133.1) | 185.1(145.9) | <0.01 |
| Total adjusted n3 fatty acid intake (mg/kg/day)[2] | 23.1(14.9) | 19(13.5) | <0.01 |
| Total adjusted n6 fatty acid intake (mg/kg/day)[2] | 204.1(116.4) | 172(100.9) | <0.01 |
| n6: n3 ratio[2] | 9.5(3.4) | 9.4(3.2) | 0.264 |

Data are the number of subjects (percentage) or mean (standard deviation).

[1]Number of participants and weighted percentage. Chi-square test was used to compare the percentage between participants with and without hyperuricemia.

[2]Weighted mean value and standard deviation (SD). Student's t-test was used to compare the mean values between participants with and without hyperuricemia. National Health and Nutrition Examination Survey (NHANES).

the ORs (95% CIs) of n3 and n6 fatty acid intake in Model 2 were 0.74 (0.58, 0.96) and 0.62 (0.50, 0.76), respectively. However, among participants aged over 60 years old, no significant association was found between dietary consumption of n3 and n6 fatty acids and the risk of hyperuricemia (Model 2). Additionally, the link between the n6:n3 ratio and the risk of hyperuricemia was not significant in any of the subgroup analyses.

## Consumption of N3 and n6 fatty acids stratified by sex

In Table 4, the correlations between dietary consumption of n3 and n6 as well as the n6:n3 ratio and the risk of hyperuricemia stratified by sex are displayed. In males, after adjusting for confounders (Model 2), the ORs (95% CIs) for hyperuricemia were 0.75 (0.63, 0.90) and 0.69 (0.59, 0.80) for consumption of n3 and n6 fatty acids, respectively. Similar associations were observed in females. However, there was no statistically significant relationship between the incidence of hyperuricemia and the n6:n3 ratio in any of the subgroup analyses.

## Sensitivity analyses

We further performed sensitivity analysis by excluding participants with obesity (BMI ≥ 30 kg/m[2]) and significant renal dysfunction (eGFR lower than 60 mL/min/1.73 m[2]). After fully adjusting for the same confounding factors, the highest tertiles for adjusted n3 and n6 intakes in all models were still negatively associated with the risk of hyperuricemia, and no significant conclusion was obtained between the n6:n3 ratio and the risk of hyperuricemia (Table S1).

## Dose-response analyses

Figs 2 and 3 depict the dose-response analyses between the consumption of n3 and n6 fatty acids and hyperuricemia. The intake of n3 fatty acids and hyperuricemia were found to be linearly correlated (P for nonlinearity = 0.0790). The prevalence of hyperuricemia decreased as the consumption of n3 fatty acids increased. Moreover, this linear inverse association between the consumption of n6 fatty acids and the prevalence of hyperuricemia was also detected (P for nonlinearity = 0.0766). Since there was no significant correlation found in the multivariable logistic regression (Model 2), a dose-response analysis between the n6:n3 ratio and hyperuricemia was not carried out.

**Table 2. Weighted odds ratios (95% confidence intervals) of hyperuricemia across tertiles of adjusted dietary n3, n6 fatty acid intake and n6: n3 ratio.**

|  | Case/Participants | Crude1 OR (95% CI) | Model 1 OR (95% CI) | Model 2 OR (95% CI) |
|---|---|---|---|---|
| **Adjusted n3 (mg/kg/day)** |  |  |  |  |
| <14.89 | 1,817/7,027 | 1.00 (Ref.) | 1.00 (Ref.) | 1.00 (Ref.) |
| 14.89 to 24.48 | 1,423/7,039 | 0.72(0.65,0.80) | 0.73(0.65,0.81) | 0.88(0.78,1.00) |
| ≥24.48 | 1,060/7,054 | 0.50(0.45,0.56) | 0.49(0.44,0.56) | 0.76(0.66,0.88) |
| **Adjusted n6 (mg/kg/day)** |  |  |  |  |
| <136.71 | 1,840/7,033 | 1.00 (Ref.) | 1.00 (Ref.) | 1.00 (Ref.) |
| 136.71 to 220.81 | 1,399/7,034 | 0.71(0.63,0.79) | 0.70(0.62,0.78) | 0.84(0.75,0.95) |
| ≥220.81 | 1,061/7,053 | 0.49(0.45,0.54) | 0.48(0.43,0.53) | 0.72(0.64,0.82) |
| **n6: n3 ratio** |  |  |  |  |
| <8.07 | 1,432/7,043 | 1.00 (Ref.) | 1.00 (Ref.) | 1.00 (Ref.) |
| 8.07 to 9.94 | 1,461/7,012 | 1.08(0.97,1.20) | 1.09(0.98,1.22) | 1.02(0.91,1.15) |
| ≥9.94 | 1,407/7,065 | 0.98(0.88,1.09) | 0.99(0.88,1.10) | 0.93(0.82,1.06) |

Model 1 adjusted for age, gender, ethnicity.

Model 2 further adjusted for educational level, marital status, annual household income, body mass index (BMI), hypertension, smoking status, work activity, recreational activity, diabetes, total cholesterol (TC), high-density lipoprotein cholesterol (HDL-C), and total triglycerides (TG). The lowest tertile of dietary n3, n6, n6: n3 intake was used as the reference group.

## Discussion

In this investigation, we indicated that the dietary consumption of n3 and n6 fatty acids was negatively correlated with the risk of hyperuricemia, while no significant association between the n6:n3 ratio and the possibility of hyperuricemia was found. Furthermore, the dose-response analyses showed linear associations between the dietary consumption of n3 and n6 fatty acids and the risk of hyperuricemia.

To the best of our knowledge, this was the first investigation into the correlation between dietary intakes of n3, n6 and the ratio of n6 to n3 fatty acids and the risk of hyperuricemia in US adults using a nationally representative sample. The results of several previous studies were consistent with our findings. An in vitro cell experiment showed that unsaturated fatty acids, especially polyunsaturated fatty acids, could strongly inhibit urate transporter 1 (URAT1), which could reabsorb urate filtered by the glomerulus of the kidney from primary urine into the blood [16]. Akhigbe et al. found that n3 fatty acids decreased testicular uric acid significantly in Torsion/Detorsion rats [15]. However, contrary to our findings, a study comprising 113 participants who were 18 years of age and older at the Hospital of the Federal University of Uberlandia revealed no correlation between n-3 and n-6 fatty acids and blood uric acid [17]. The disparity could be explained by a number of things. First, when compared to that of our study (21,120 American adults), the sample of the above research was too small (113 American adults). Second, the participants in the earlier study were all kidney transplant patients, and their uric acid metabolism was abnormal because of kidney dysfunction.

Although the underlying mechanisms between dietary consumption of n3 and n6 fatty acids and the risk of hyperuricemia are not completely understood, several possibilities have been proposed. First, prior research revealed that n3 and n6 fatty acids were inversely connected to the incidence of hypertension and had a protective effect on defective endothelial cells and nitric oxide-dependent responses [22,23]. Meanwhile, hypertension could elevate serum uric acid levels by increasing renal vascular resistance and uric acid retention [24].

**Table 3. The ORs (95%CIs) of hyperuricemia by adjusted dietary n3, n6 fatty acid intake and n6: n3 ratio.**

| | 20 ≤ Age < 40 Years (n = 6,756) | | | 40 ≤ Age < 60 Years (n = 7,196) | | | Age ≥ 60 Years (n = 7,168) | | |
|---|---|---|---|---|---|---|---|---|---|
| | Crude OR (95% CI) | Model 1 OR (95% CI) | Model 2 OR (95% CI) | Crude OR (95% CI) | Model 1 OR (95% CI) | Model 2 OR (95% CI) | Crude OR (95% CI) | Model 1 OR (95% CI) | Model 2 OR (95% CI) |
| **Adjusted n3 (mg/kg/day)** | | | | | | | | | |
| <14.89 | 1.00 (Ref.) | 1.00 (Ref.) | 1.00 (Ref.) | 1.00 (Ref.) | 1.00 (Ref.) | 1.00 (Ref.) | 1.00 (Ref.) | 1.00 (Ref.) | 1.00 (Ref.) |
| 14.89 to 24.48 | 0.77 | 0.70 | 0.91 | 0.71 | 0.70 | 0.82 | 0.73 | 0.73 | 0.87 |
| | (0.65,0.90) | (0.60,0.82) | (0.75,1.09) | (0.58,0.88) | (0.57,0.86) | (0.65,1.04) | (0.61,0.87) | (0.61,0.88) | (0.72,1.06) |
| ≥24.48 | 0.40 | 0.35 | 0.61 | 0.53 | 0.51 | 0.74 | 0.59 | 0.60 | 0.88 |
| | (0.33,0.48) | (0.29,0.43) | (0.48,0.77) | (0.42,0.66) | (0.41,0.64) | (0.58,0.96) | (0.50,0.70) | (0.50,0.71) | (0.71,1.09) |
| **Adjusted n6 (mg/kg/day)** | | | | | | | | | |
| <136.71 | 1.00 (Ref.) | 1.00 (Ref.) | 1.00 (Ref.) | 1.00 (Ref.) | 1.00 (Ref.) | 1.00 (Ref.) | 1.00 (Ref.) | 1.00 (Ref.) | 1.00 (Ref.) |
| 136.71 to 220.81 | 0.74 | 0.67 | 0.90 | 0.70 | 0.68 | 0.78 | 0.71 | 0.72 | 0.85 |
| | (0.62,0.88) | (0.56,0.81) | (0.73,1.10) | (0.58,0.84) | (0.57,0.82) | (0.64,0.96) | (0.59,0.86) | (0.60,0.88) | (0.70,1.04) |
| ≥220.81 | 0.45 | 0.38 | 0.68 | 0.49 | 0.46 | 0.62 | 0.58 | 0.59 | 0.86 |
| | (0.38,0.54) | (0.32,0.46) | (0.56,0.84) | (0.42,0.59) | (0.39,0.55) | (0.50,0.76) | (0.50,0.69) | (0.51,0.70) | (0.70,1.05) |
| **n6: n3 ratio** | | | | | | | | | |
| <8.07 | 1.00 (Ref.) | 1.00 (Ref.) | 1.00 (Ref.) | 1.00 (Ref.) | 1.00 (Ref.) | 1.00 (Ref.) | 1.00 (Ref.) | 1.00 (Ref.) | 1.00 (Ref.) |
| 8.07 to 9.94 | 1.03 | 1.00 | 0.98 | 1.11 | 1.08 | 1.05 | 1.18 | 1.19 | 1.02 |
| | (0.85,1.25) | (0.82,1.23) | (0.78,1.23) | (0.89,1.37) | (0.87,1.34) | (0.83,1.32) | (1.00,1.41) | (1.00,1.41) | (0.85,1.23) |
| ≥9.94 | 1.08 | 1.02 | 1.00 | 0.99 | 0.94 | 0.89 | 1.05 | 1.05 | 0.97 |
| | (0.88,1.32) | (0.83,1.25) | (0.82,1.22) | (0.80,1.23) | (0.75,1.17) | (0.69,1.14) | (0.88,1.26) | (0.88,1.26) | (0.80,1.19) |

Model 1 adjusted for age, gender, ethnicity.

Model 2 further adjusted for educational level, marital status, annual household income, body mass index (BMI), hypertension, smoking status, work activity, recreational activity, diabetes, total cholesterol (TC), high-density lipoprotein cholesterol (HDL-C), and total triglycerides (TG). The lowest tertile of dietary n3, n6, n6: n3 intake was used as the reference group.

Thus, n3 and n6 fatty acids might decrease susceptibility to hyperuricemia by lowering blood pressure. Second, polyunsaturated fatty acids could prevent and reverse insulin resistance [25,26]. Insulin resistance increased serum uric acid levels by damaging urinary endothelial function and reducing uric acid excretion [27,28]. Therefore, n3 and n6 fatty acids were likely inversely related to elevated serum uric acid levels because of their function of improving insulin resistance. To understand the mechanisms behind the link between dietary consumption of n-3 and n-6 fatty acids and the risk of hyperuricemia, more research is needed.

Previous research revealed that the intake of dairy products [29–31], vitamin C [32], vitamin D [33], soy products [34], dietary fiber [35], magnesium [36], and dietary zinc intake [37] were adversely connected to the incidence of hyperuricemia. Thus, dietary changes may be connected to hyperuricemia. Our findings showed that increased dietary polyunsaturated fatty acids might decrease the risk of hyperuricemia. In addition, green leafy vegetables, legumes, nuts, and seafood are all rich in polyunsaturated fatty acids [11,38], and these foods are common in our daily meals. It is beneficial for individuals to maintain sufficient polyunsaturated fatty acid intake to prevent the risk of hyperuricemia.

When stratified by age, the dietary consumption of n3 and n6 fatty acids was associated with a nonsignificant lower risk of hyperuricemia among people aged 60 years old and above in the fully adjusted model. This may be explained by the high prevalence of hypertension among older adults. Some antihypertensive medications, such as loop and thiazide diuretics, cause relative

**Table 4.** The ORs (95%CIs) of hyperuricemia by adjusted dietary n3, n6 fatty acid intake and n6: n3 ratio, stratified by gender.

| | Men (n = 10,222) | | | Women (n = 10,898) | | |
|---|---|---|---|---|---|---|
| | Crude OR (95% CI) | Model 1 OR (95% CI) | Model 2 OR (95% CI) | Crude OR (95% CI) | Model 1 OR (95% CI) | Model 2 OR (95% CI) |
| **Adjusted n3 (mg/kg/day)** | | | | | | |
| <14.89 | 1.00 (Ref.) | 1.00 (Ref.) | 1.00 (Ref.) | 1.00 (Ref.) | 1.00 (Ref.) | 1.00 (Ref.) |
| 14.89 to 24.48 | 0.79(0.69,0.90) | 0.79(0.69,0.91) | 0.93(0.79,1.10) | 0.64(0.57,0.72) | 0.64(0.56,0.72) | 0.79(0.69,0.92) |
| ≥24.48 | 0.53(0.45,0.61) | 0.53(0.45,0.61) | 0.75(0.63,0.90) | 0.45(0.37,0.54) | 0.44(0.36,0.54) | 0.73(0.59,0.90) |
| **Adjusted n6 (mg/kg/day)** | | | | | | |
| <136.71 | 1.00 (Ref.) | 1.00 (Ref.) | 1.00 (Ref.) | 1.00 (Ref.) | 1.00 (Ref.) | 1.00 (Ref.) |
| 136.71 to 220.81 | 0.68(0.58,0.81) | 0.68(0.57,0.80) | 0.78(0.65,0.93) | 0.70(0.61,0.82) | 0.72(0.62,0.83) | 0.90(0.76,1.06) |
| ≥220.81 | 0.51(0.45,0.58) | 0.50(0.44,0.58) | 0.69(0.59,0.80) | 0.42(0.35,0.51) | 0.43(0.36,0.52) | 0.71(0.58,0.87) |
| **n6: n3 ratio** | | | | | | |
| <8.07 | 1.00 (Ref.) | 1.00 (Ref.) | 1.00 (Ref.) | 1.00 (Ref.) | 1.00 (Ref.) | 1.00 (Ref.) |
| 8.07 to 9.94 | 1.15(0.97,1.36) | 1.14(0.96,1.35) | 1.07(0.89,1.28) | 0.99(0.87,1.13) | 1.05(0.91,1.21) | 0.95(0.80,1.13) |
| ≥9.94 | 1.04(0.89,1.22) | 1.02(0.87,1.20) | 0.97(0.81,1.15) | 0.88(0.74,1.04) | 0.97(0.81,1.18) | 0.91(0.73,1.13) |

Model 1 adjusted for age, gender, ethnicity.

Model 2 further adjusted for educational level, marital status, annual household income, body mass index (BMI), hypertension, smoking status, work activity, recreational activity, diabetes, total cholesterol (TC), high-density lipoprotein cholesterol (HDL-C), and total triglycerides (TG). The lowest tertile of dietary n3, n6, n6: n3 intake was used as the reference group.

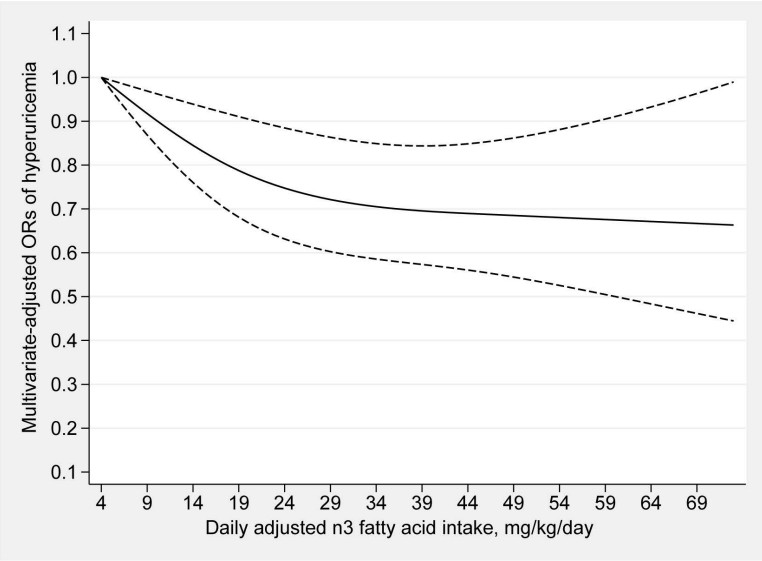

**Fig 2. Dose–response relationship between n3 fatty acids intake and the risk of hyperuricemia.** The solid line represents the OR values and dashed lines rep-resent the 95% confidence intervals.

hypovolemia, which induces a compensatory increase in proximal uric acid reabsorption [39], and approximately two-thirds of all uric acid is excreted by the kidneys [28].

Our study has several strengths. First, this was the first study to investigate the correlation between dietary n3 and n6 fatty acids and the susceptibility of hyperuricemia with their

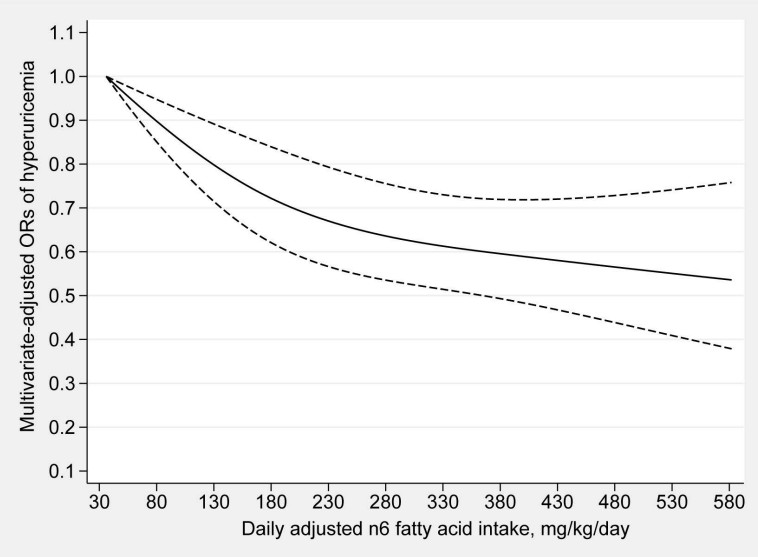

**Fig 3. Dose–response relationship between n6 fatty acids intake and the risk of hyperuricemia.** The solid line represents the OR values and dashed lines rep-resent the 95% confidence intervals.

dose-response relationship among US adults using the largest nationally representative sample (21,120 participants). Second, to lessen the impact of visceral fat accumulation and kidney on serum uric acid, we performed sensitivity analyses by removing subjects with obesity (BMI 30 kg/m2) and substantial renal dysfunction (eGFR lower than 60 mL/min/1.73 m2) [40,41]. Third, the utilization of trained personnel who adhered to standardized protocols for measuring essential study data and conducting interviews enhanced the accuracy and efficiency of the data collection process.

Additionally, our study has some flaws. First, it is impossible to determine causality due to the cross-sectional design. Future studies utilizing longitudinal designs are necessary to confirm our findings. Second, two 24-hour food recall interviews were used to collect the dietary data, which may not have provided an accurate representation of long-term average intake. Third, our study did not take into account the intake of n3 and n6 fatty acids from dietary supplements, which could be an important source of these nutrients. Fourth, our study population was limited to Americans, and it is unclear whether our results can be generalized to other countries.

## Conclusion

In conclusion, the findings of this study indicated that the dietary consumption of n3 and n6 fatty acids was inversely correlated with hyperuricemia in US adults, independent of some major confounding factors. The n6:n3 ratio was not shown to be significantly correlated with hyperuricemia. To support our findings, additional large-scale prospective studies using more precise dietary survey techniques are needed.

## Supporting information

**Table S1. The ORs (95% CIs) for hyperuricemia by adjusted dietary n3, n6 fatty acid intake and n6: n3 ratio excluding participants with eGFR lower than 60 mL/min/1.73 m2 and BMI ≥ 30 kg/m2.**
(DOCX)

**Data S1. n3-n6.Data**.
(ZIP)

**File S1. Code.**
(ZIP)

## Author contributions

**Conceptualization:** Huakai Wang, Chao Zhang, Yuxin Sun.

**Data curation:** Huakai Wang.

**Formal analysis:** Chao Zhang, Yuxin Sun, Zhe Wang.

**Funding acquisition:** Huakai Wang.

**Supervision:** Honggang Xiang.

**Writing – original draft:** Huakai Wang, Sirui Sun.

**Writing – review & editing:** Honggang Xiang.

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
