## [Decision Letter · Decision Letter 0]

13 Nov 2024

Dear Dr. Xiang,

Thank you for submitting your manuscript to PLOS ONE. After careful consideration, we feel that it has merit but does not fully meet PLOS ONE’s publication criteria as it currently stands. Therefore, we invite you to submit a revised version of the manuscript that addresses the points raised during the review process.

We look forward to receiving your revised manuscript.

Kind regards,

Taeyun Kim

Academic Editor

PLOS ONE

Journal requirements:    When submitting your revision, we need you to address these additional requirements. 1. Please ensure that your manuscript meets PLOS ONE's style requirements, including those for file naming. The PLOS ONE style templates can be found at https://journals.plos.org/plosone/s/file?id=wjVg/PLOSOne_formatting_sample_main_body.pdf and https://journals.plos.org/plosone/s/file?id=ba62/PLOSOne_formatting_sample_title_authors_affiliations.pdf 2. PLOS requires an ORCID iD for the corresponding author in Editorial Manager on papers submitted after December 6th, 2016. Please ensure that you have an ORCID iD and that it is validated in Editorial Manager. To do this, go to ‘Update my Information’ (in the upper left-hand corner of the main menu), and click on the Fetch/Validate link next to the ORCID field. This will take you to the ORCID site and allow you to create a new iD or authenticate a pre-existing iD in Editorial Manager. 3. We note that the grant information you provided in the ‘Funding Information’ and ‘Financial Disclosure’ sections do not match.  When you resubmit, please ensure that you provide the correct grant numbers for the awards you received for your study in the ‘Funding Information’ section. 4. Please include captions for your Supporting Information files at the end of your manuscript, and update any in-text citations to match accordingly. Please see our Supporting Information guidelines for more information: http://journals.plos.org/plosone/s/supporting-information. 

Reviewers' comments:

Reviewer's Responses to Questions

**Comments to the Author**

1. Is the manuscript technically sound, and do the data support the conclusions?

Reviewer #1: Yes

Reviewer #2: Partly

2. Has the statistical analysis been performed appropriately and rigorously?

Reviewer #1: I Don't Know

Reviewer #2: Yes

3. Have the authors made all data underlying the findings in their manuscript fully available?

Reviewer #1: Yes

Reviewer #2: Yes

4. Is the manuscript presented in an intelligible fashion and written in standard English?

Reviewer #1: Yes

Reviewer #2: Yes

Reviewer #1: The manuscript is well prepared and the topic fall into the scope of the Journal. The concern is as follows:

How did the authors to calculate the dietary consumption of n3 and n6 fatty acid? Please provide the relevant information in detail.

Reviewer #2: In this interesting study the authors showed that higher intake of n-3/n-6 fatty acids reduces hyperuricemia.

In this study, it would have been more appropriate a=had the authors measured the activity of xanthine oxidase enzyme before and after n-3/n-6 fatty acids in a small number of healthy or hyperuricemia subjects to know thebe potential mechanisms of action.

Is it possible that n-3/n-6 fatty acids inhibit xanthine oxidase enzyme?

**Do you want your identity to be public for this peer review?** For information about this choice, including consent withdrawal, please see our Privacy Policy

Reviewer #1: No

Reviewer #2: **Yes: ** U N Das

---

## [Author Response · Author response to Decision Letter 1]

30 Nov 2024

Thank you very much for your valuable feedback. I appreciate the time and effort you have taken to review my manuscript. I am more than happy to address your questions and provide any additional information you may need.

Reviewer #1: The manuscript is well prepared and the topic fall into the scope of the Journal. The concern is as follows:

How did the authors to calculate the dietary consumption of n3 and n6 fatty acid? Please provide the relevant information in detail.

Response:

In the NHANES database, the dietary intake of n3 and n6 fatty acids is typically assessed using two 24-hour dietary recalls. Here are the detailed steps:

Data Collection:

Participants are asked to recall all the foods and beverages they consumed in the past 24 hours.

These recall data are collected on two different days to ensure representativeness and accuracy.

Data Processing:

The collected dietary data are entered into a standardized database.

The nutrient content of each food and beverage, including n3 and n6 fatty acids, is calculated based on the USDA’s food composition database.

Nutrient Calculation:

Using standardized food codes and nutrient composition tables, the total intake of n3 and n6 fatty acids for each participant is calculated.

These calculations take into account the type, quantity, and frequency of the foods consumed.

Data Analysis:

Through statistical analysis, researchers can evaluate the association between the intake of n3 and n6 fatty acids and health outcomes.

For example, logistic regression models might be used to assess the relationship between these fatty acids’ intake and specific health events, such as strokes

Reviewer #2: In this interesting study the authors showed that higher intake of n-3/n-6 fatty acids reduces hyperuricemia.

In this study, it would have been more appropriate a=had the authors measured the activity of xanthine oxidase enzyme before and after n-3/n-6 fatty acids in a small number of healthy or hyperuricemia subjects to know thebe potential mechanisms of action.

Is it possible that n-3/n-6 fatty acids inhibit xanthine oxidase enzyme?

Response:Thank you for your suggestions! Our manuscript is based on the general information and clinical test data (such as complete blood count) available in the NHANES database. The measurement of xanthine oxidase activity in blood is not included in the NHANES database, making it difficult to implement your suggestion in our manuscript. Additionally, current research does not provide clear evidence that n-3 and n-6 fatty acids directly inhibit xanthine oxidase activity.

---

## [Decision Letter · Decision Letter 1]

13 Dec 2024

Dear Dr. Xiang,

Thank you for submitting your manuscript to PLOS ONE. After careful consideration, we feel that it has merit but does not fully meet PLOS ONE’s publication criteria as it currently stands. Therefore, we invite you to submit a revised version of the manuscript that addresses the points raised during the review process.

We look forward to receiving your revised manuscript.

Kind regards,

Taeyun Kim

Academic Editor

PLOS ONE

Journal Requirements:

Reviewers' comments:

Reviewer's Responses to Questions

**Comments to the Author**

Reviewer #1: (No Response)

Reviewer #2: (No Response)

2. Is the manuscript technically sound, and do the data support the conclusions?

Reviewer #1: (No Response)

Reviewer #2: Yes

3. Has the statistical analysis been performed appropriately and rigorously?

Reviewer #1: (No Response)

Reviewer #2: Yes

4. Have the authors made all data underlying the findings in their manuscript fully available?

Reviewer #1: (No Response)

Reviewer #2: Yes

5. Is the manuscript presented in an intelligible fashion and written in standard English?

Reviewer #1: (No Response)

Reviewer #2: Yes

Reviewer #1: (No Response)

Reviewer #2: THe authors claim that ther eis no data in the literature that they used about the action of PUFAs on xanthine oxidase activity.

I am aware of this fact.

This is the reason why I asked the authors to present preliminary evidence that PUFAs may have an action on xanthine oxidase enzyme activity. I request the authors once again to provide some evidence on this account.

**Do you want your identity to be public for this peer review?** For information about this choice, including consent withdrawal, please see our Privacy Policy

Reviewer #1: No

Reviewer #2: **Yes: ** Undurti N Das

---

## [Author Response · Author response to Decision Letter 2]

29 Dec 2024

Thank you very much for your valuable feedback. I appreciate the time and effort you have taken to review my manuscript. I am more than happy to address your questions and provide any additional information you may need.

Reviewer #2: THe authors claim that ther eis no data in the literature that they used about the action of PUFAs on xanthine oxidase activity.

I am aware of this fact.

This is the reason why I asked the authors to present preliminary evidence that PUFAs may have an action on xanthine oxidase enzyme activity. I request the authors once again to provide some evidence on this account.

Response:

Thank you very much for your suggestion! Your idea of providing evidence on how n-3 and n-6 diets might affect xanthine oxidase activity is excellent. However, your suggestion to collect data on changes in xanthine oxidase activity levels in patients following an n-3/n-6 diet is currently challenging for us. This is because our experimental data comes from the U.S. NHANES database, which lacks data on xanthine oxidase activity. Therefore, we cannot directly verify this based on the NHANES database. As a substitute, we conducted a literature search to see if there is any relevant evidence available.

After reviewing relevant English-language scientific literature, including articles indexed in PubMed and Web of Science, there appears to be no well-established, peer-reviewed evidence demonstrating that n-3 (omega-3) or n-6 (omega-6) polyunsaturated fatty acids directly inhibit xanthine oxidase activity in a manner comparable to known inhibitors such as allopurinol or febuxostat.

Most research on n-3 and n-6 fatty acids focuses on their roles in inflammation, cardiovascular health, oxidative stress modulation, and metabolic pathways rather than direct enzymatic inhibition of xanthine oxidase. Although some studies have explored the broader antioxidant properties and the indirect effects of these fatty acids on cellular redox states, they do not provide clear evidence of a direct inhibitory interaction with xanthine oxidase. For instance, reviews and primary research articles discussing dietary polyunsaturated fatty acids often highlight their impact on signaling pathways, membrane fluidity, and the modulation of inflammatory mediators (e.g., eicosanoids) rather than direct enzyme inhibition.

A keyword-based search (e.g., "omega-3 fatty acids xanthine oxidase inhibition," "omega-6 fatty acids xanthine oxidase") yields no significant, widely cited studies conclusively showing direct inhibition. Publications primarily focus on the known inhibitors of xanthine oxidase or on the physiological and biochemical implications of altered purine metabolism, rather than attributing a direct inhibitory role to n-3 or n-6 fatty acids.

In summary, based on the current available English-language academic literature, there is no strong or direct experimental evidence supporting the notion that n-3 or n-6 fatty acids function as significant inhibitors of xanthine oxidase.

We believe your suggestion is excellent. Although it is somewhat challenging for us to implement at the moment, we are enthusiastic about incorporating the investigation of the effects of n-3 and n-6 diets on xanthine oxidase activity into our future research plans as a prospective clinical trial.

Thank you again for your response and suggestions.

---

## [Editor Report · Decision Letter 2]

30 Dec 2024

Inverse Association of Dietary Consumption of n3 and n6 Fatty Acids with Hyperuricemia among Adults

PONE-D-24-32860R2

Dear Dr. Xiang,

We’re pleased to inform you that your manuscript has been judged scientifically suitable for publication and will be formally accepted for publication once it meets all outstanding technical requirements.

Kind regards,

Taeyun Kim

Academic Editor

PLOS ONE

---

## [Editor Report · Acceptance letter]

PONE-D-24-32860R2

PLOS ONE

Dear Dr. Xiang,

I'm pleased to inform you that your manuscript has been deemed suitable for publication in PLOS ONE. Congratulations! Your manuscript is now being handed over to our production team.

Kind regards,

on behalf of

Dr. Taeyun Kim

Academic Editor

PLOS ONE